# Developing a Mathematical Model of Intracellular Calcium Dynamics for Evaluating Combined Anticancer Effects of Afatinib and RP4010 in Esophageal Cancer

**DOI:** 10.3390/ijms23031763

**Published:** 2022-02-03

**Authors:** Yan Chang, Marah Funk, Souvik Roy, Elizabeth Stephenson, Sangyong Choi, Hristo V. Kojouharov, Benito Chen, Zui Pan

**Affiliations:** 1College of Nursing and Health Innovation, The University of Texas at Arlington, Arlington, TX 76019, USA; yan.chang@uta.edu (Y.C.); sangyong.choi@uconn.edu (S.C.); 2Department of Mathematics, The University of Texas at Arlington, Arlington, TX 76019, USA; marah.townzen@mavs.uta.edu (M.F.); souvik.roy@uta.edu (S.R.); elizabeth.stephenson@mavs.uta.edu (E.S.); bmchen@uta.edu (B.C.); 3Institute of Science and Technology Austria, Am Campus 1, 3400 Klosterneuburg, Austria; 4Department of Nutritional Sciences, University of Connecticut, Mansfield, CT 06269, USA

**Keywords:** Orai1, store-operated Ca^2+^ entry, tyrosine kinase inhibitor, esophageal cancer, fluxes equations

## Abstract

Targeting dysregulated Ca^2+^ signaling in cancer cells is an emerging chemotherapy approach. We previously reported that store-operated Ca^2+^ entry (SOCE) blockers, such as RP4010, are promising antitumor drugs for esophageal cancer. As a tyrosine kinase inhibitor (TKI), afatinib received FDA approval to be used in targeted therapy for patients with EGFR mutation-positive cancers. While preclinical studies and clinical trials have shown that afatinib has benefits for esophageal cancer patients, it is not known whether a combination of afatinib and RP4010 could achieve better anticancer effects. Since TKI can alter intracellular Ca^2+^ dynamics through EGFR/phospholipase C-γ pathway, in this study, we evaluated the inhibitory effect of afatinib and RP4010 on intracellular Ca^2+^ oscillations in KYSE-150, a human esophageal squamous cell carcinoma cell line, using both experimental and mathematical simulations. Our mathematical simulation of Ca^2+^ oscillations could fit well with experimental data responding to afatinib or RP4010, both separately or in combination. Guided by simulation, we were able to identify a proper ratio of afatinib and RP4010 for combined treatment, and such a combination presented synergistic anticancer-effect evidence by experimental measurement of intracellular Ca^2+^ and cell proliferation. This intracellular Ca^2+^ dynamic-based mathematical simulation approach could be useful for a rapid and cost-effective evaluation of combined targeting therapy drugs.

## 1. Introduction

Esophageal cancer (EC) is the sixth leading cause of cancer mortality worldwide [1]. In 2021, in the United States alone, there is an estimated number of 19,260 new cases of EC and 15,530 deaths from EC (www.seer.cancer.gov/statfacts, accessed on 24 January 2022). Two main types of EC are distinguished by different etiological and pathological characteristics: esophageal squamous cell carcinoma (ESCC) and adenocarcinoma (EAC). While EAC is more prevalent in the USA, ESCC predominates among Asians and male African Americans [2]. Both cancers remain asymptomatic; therefore, patients are usually diagnosed at relatively late stages, with an overall 5-year survival rate of below 20%, according to the SEER Cancer Statistics review [3]. The need for development of accurate and timely treatments for EC, thus, is crucial. In recent years, targeting dysregulated Ca^2+^ signaling in cancer cells has become an active research area to develop new chemotherapy drugs [4].

Ca^2+^ signaling plays an important role in cell proliferation, apoptosis, autophagy, migration and cell cycle; thus, its dysregulation is associated with tumor initiation, angiogenesis, progression and metastasis [4]. The intracellular Ca^2+^ signals have different forms, such as Ca^2+^ spikes, waves and oscillations [5,6]. They are regulated by both intracellular Ca^2+^ release and extracellular Ca^2+^ influx [7]. Store-operated Ca^2+^ entry (SOCE) is a ubiquitous important extracellular Ca^2+^ influx, which is mainly mediated by two proteins, i.e., stromal-interacting molecule 1 (STIM1) as endoplasmic reticulum (ER) Ca^2+^ storage sensor and Orai1 as plasma membrane (PM) Ca^2+^ channel. During activation of SOCE, the depletion of ER Ca^2+^ stores triggers translocation of STIM1 to ER-PM junctions, followed by binding and activation of Orai1 channel at PM [8]. Accumulating evidence has shown that STIM1/Orai1-mediated SOCE actively participates in the progression of many cancers [9], such as breast cancer [10], pancreatic adenocarcinoma [11] and prostate cancer [12]. We previously reported that high expression of Orai1 in tumor tissues is associated with poor prognosis in ESCC patients, and SOCE-mediated intracellular Ca^2+^ oscillations regulate cell proliferation, migration and invasion in ESCC cells [13]. Decreased Orai1-mediated SOCE, either by gene knockdown or pharmacological channel blockers, is able to reduce the frequency of intracellular Ca^2+^ oscillations in cultured ESCC cells and to inhibit tumor growth in preclinical animal models. Among many SOCE blockers, RP4010 is a recently developed one in clinical trial phase I/IB. Our published study suggested that RP4010 is a promising chemotherapy drug targeting SOCE for ESCC patients [14]. 

The overexpression and/or mutations of epidermal growth factor receptor (EGFR) family proteins, i.e., EGFR (HER1), ERBB2, ERBB3 and ERBB4, are often found in multiple types of cancer cells and are considered to be a significant prognostic indicator in the clinical intervention of cancer. Upon binding with the EGF or other ligands, EGFR undergoes a conformational change on the extracellular domain, followed by dimerization and trans-phosphorylation of tyrosine kinases in the intracellular domain [15]. Activated EGFR tyrosine kinase triggers multiple signaling pathways. One is through phosphorylation of phospholipase C-γ (PLC-γ)/inositol-1,4,5-trisphosphate (IP3)/IP3 receptor (IP3R) pathway and results in intracellular Ca^2+^ release from ER [16]. Others include downstream PI3K/AKT and MEK/ERK signaling pathways, which are essential for cancer progression. Accordingly, many tyrosine kinase inhibitors (TKIs) have been developed, and more than 20 have received FDA approval as targeting cancer therapy for head and neck, lung, breast and colon cancers [17]. Afatinib is one of the second-generation TKIs and has an irreversible inhibition on both EGFR and HER2 [18]. As its regulatory approval for use as a treatment for non-small-cell lung cancer and squamous cell carcinoma of the lung, many preclinical and clinical studies support its benefit for patients with other cancers, including recurrent and/or metastatic ESCC [19]. However, a challenge of using afatinib, as well as other TKIs, for chemotherapy is that most patients show TKI resistance after initial response [20]. Consequently, combinational treatments of chemotherapy, immunotherapy or other kinds of targeting therapy have been proposed to improve the treatment effect and patient survival rate. Considering that EGFR stimulates intracellular Ca^2+^ release and SOCE-mediated Ca^2+^ influx regulates EGFR downstream AKT and ERK activity, we hypothesize that combined afatinib and RP4010 may achieve an enhanced anticancer effect in ESCC cells. Since the dynamic of intracellular Ca^2+^ is at the intersection between the two signaling pathways and it can be rapidly simulated by a mathematical model, this study used both mathematical simulation and experimental data of the dynamics of intracellular Ca^2+^ in cultured ESCC cell lines to evaluate the combined anticancer effects of afatinib and RP4010.

## 2. Results

### 2.1. Inhibited Intracellular Ca^2+^ Oscillations by Afatinib in ESCC Cells

We extracted RNA-Seq data from the TCGA database to compare the expression of EGFR in tumor tissues removed from ESCC patients with normal human esophageal tissues (Figure 1a). The average expression levels, presented as transcripts per million (TMP) in tumor and non-tumor tissues, were 38.4 and 20.4, respectively, suggesting upregulated expression of EGFR in ESCC tumor tissues. Interestingly, more than 16% ESCC patients showed an increased amplification of EGFR, whereas mutation was much less common. In order to reveal the expression of individual EGFR family proteins in ESCC cells, Western blot was employed by using specific antibodies against each EGFR family protein. We compared their expression in four human ESCC cell lines (KYSE-30, KYSE-150, KYSE-70 and KYSE-790) and a non-tumorous esophageal epithelial cell line (Het-1A). All four proteins, i.e., EGFR, ERBB2, ERBB3 and ERBB4, were expressed in ESCC cells, but only EGFR and ERBB2 were significantly upregulated in ESCC cells compared to those in Het-1A cells (Figure 1b,c). As described earlier, afatinib is an effective TKI for EGFR/ERBB2; therefore, we suspected that these ESCC cells should be sensitive to afatinib. The KYSE-150 cell line was selected as a representative ESCC cell model in the following studies.

Our previous study suggested intracellular Ca^2+^ oscillations as an important Ca^2+^ code for activation of downstream signaling pathways in ESCC cells [13,14]. Since EGFR regulates Ca^2+^ release through PLC-γ/IP3/IP3R axis, we next tested whether afatinib could impair intracellular Ca^2+^ oscillations. KYSE-150 cells were loaded with Fluo-4 AM, a fluorescent Ca^2+^ indicator, followed by time-lapse live-cell imaging. More than half of the cells underwent intracellular Ca^2+^ oscillations, as indicated by spontaneous fluctuation in fluorescence (Figure 1c, upper panel, control). However, treatment of 5 μM afatinib for 4 h clearly impaired intracellular Ca^2+^ oscillations in KYSE-150 cells, as evidenced by almost no spontaneous fluctuation in fluorescence within the 5 min recording window (Figure 1d, lower panel, afatinib).

### 2.2. Decreased Frequency of Intracellular Ca^2+^ Oscillations by Treatment of RP4010 or Afatinib 

The previous studies from our and other groups have shown that the frequency is the essential parameter of intracellular Ca^2+^ oscillations to regulate downstream signaling pathways [13]; therefore, we evaluated the impact of RP4010 or afatinib on the frequency of Ca^2+^ oscillations in KYSE-150 cells. Following the same procedure as described earlier, we performed live-cell Ca^2+^ imaging in KYSE-150 cells treated with different concentrations of afatinib or RP4010 (Table 1 and Figure 2). The fluorescent intensity changes in each individual cell, reflecting intracellular Ca^2+^oscillations, were extracted as ΔF/F_0_ (representative traces in Figure 2b–d). The period between the two peaks of the ΔF/F_0_ curve was calculated as T (second) in each group, and frequency (Hz) was calculated as f = 1/T. KYSE-150 cells had active intracellular Ca^2+^ oscillations with a frequency of 0.030 Hz (Figure 2b). Treatment of either RP4010 or afatinib significantly decreased the frequency of intracellular Ca^2+^ oscillations in KYSE-150 cells (Figure 2c,d). Interestingly, both RP4010 and afatinib affected the periods of Ca^2+^ oscillations in a dose-dependent manner (Table 1). Treatment of 10 µM RP4010 and 5 µM afatinib extended the period of intracellular Ca^2+^ oscillations in KYSE-150 cells from 32.8 s to 122.1 and 73.4 s, respectively. Similar results were observed in another ESCC cell line (Appendix A).

### 2.3. Mathematical Simulation of Intracellular Ca^2+^ Oscillations 

We simulated the dynamics of [Ca^2+^]_cyto_ in KYSE-150 cells by using a mathematical model, as described in “Mathematical Modeling”. The parameter values in Table 2 were taken from the model presented by Sneyd et al. [6], with the exceptions of kf (IP3/IP3R-mediated ER Ca^2+^ release) and α1 (SOCE-mediated Ca^2+^ influx). By adjusting the key parameters for α1, *k_f_* and SERCA pump-mediated ER Ca^2+^ uptake (Vserca) (Table 2), we were able to simulate the oscillating patterns of [Ca^2+^]_cyto_ in KYSE-150 cells with the period at 32.8 s (Figure 3a, green trace). Our previous report showed that non-tumorous esophageal epithelial cells contain less active SOCE and ER Ca^2+^ release [13]. Accordingly, we adjusted the values of α1 and *k_f_* by dividing those numbers in cancer cells by 3 and 5, respectively. We were able to obtain extended period of intracellular Ca^2+^ oscillations at 600.9 s in Het-1A cells (Figure 3a, blue trace); this is consistent with our previous report of quiescent intracellular Ca^2+^ oscillations in these cells. The Michaelis–Menten kinetics parameters, *c_i_* and *k_i_*, were determined by fitting to experimental data relating to the period of Ca^2+^ oscillations to a given dose of one or both drugs. 

We then used a global sensitivity analysis, using a multidimensional parameter space to assess the effects of uncertainties of the drug parameters on the period of oscillation. For this purpose, we used a joint Latin hypercube sampling (LHS) method and partial rank correlation coefficient (PRCC) analysis (LHS-PRCC). In this method, we started with the eight important uncertain parameters, namely kf, α1, c1, c2, c3, k1, k2 and k3, which correspond to the dynamics of the SOCE channels and the drug interactions. We then used a Monte Carlo simulation technique to generate M random numbers (using a Weibull distribution) for each of the parameters. Each such set of numbers corresponded to a parameter vector that is used to compute the period of oscillation, and, thus, we obtained M periods of oscillation. We finally computed the PRCC and the *p*-values (using the Student’s *t*-statistic) to ascertain the most sensitive parameters that correspond to high PRCC values or *p*-values < 0.05. The detailed steps of the LHS-PRCC algorithm for sensitivity analysis can be found in [21,22]. The baseline parameter values are listed in Table 3, together with a brief summary of the *p*-values and the PRCC coefficients. 

From Table 3, we can see that, except for kf, c1 and k1, the other five parameters are the sensitive ones that affect the period of oscillation. Out of the sensitive parameters, α1 and c2 are the most sensitive ones. We also note that the corresponding PRCC values for α1 and c2 are −0.831 and 0.812, respectively. This implies that, while an increase in α1 is responsible for the decrease in the period of oscillation, the opposite holds for c1.

Next, we simulated the impact of RP4010 or afatinib on [Ca^2+^]_cyto_ by introducing variables in the computational model, as described in “Addition of TKI and SOCE Blocker in the Computational Model”. The simulated periods of intracellular Ca^2+^ oscillations also presented the dose-dependent inhibitory effect of RP410 or afatinib. For example, when 0.5 or 3.0 µM RP4010 was included in the computational simulation, the periods were extended to 46.3 and 83.8 s, respectively (Figure 3b,c). When 0.5 µM afatinib was included, the periods were extended to 47.4 s. The simulated periods for each compound fitted well with the experimental data (Table 1; and Figure 4c, red and blue curves).

### 2.4. Effects of Combined RP4010 and Afatinib on Intracellular Ca^2+^ Oscillations 

We then simulated [Ca^2+^]_cyto_, including both RP4010 and afatinib, into the computational model. Based on the dose-dependent curve for each individual drug, we selected a molar ratio of 2:1 for RP4010 and afatinib for the following studies. As shown in Figure 4a, the computer simulated [Ca^2+^]_cyto_ displayed oscillations with a period of 92.8 s when 2 µM RP4010 + 1 µM afatinib was included into the calculation. To validate the predicted period of intracellular Ca^2+^ oscillations, we conducted live-cell intracellular Ca^2+^ measurements in KYSE-150 cells treated with combined 2 µM RP4010 + 1 µM afatinib (Figure 4b). The calculated period of intracellular Ca^2+^ oscillations from experimental data was 89.4 s, suggesting that the mathematical model could predict the intracellular Ca^2+^ oscillations in cells treated with two different drugs. The periods of intracellular Ca^2+^ oscillations in cells treated with either 2 µM RP4010 or 1 µM afatinib alone were recorded as 56.5 and 52.7 s, respectively (Table 1). The combined treatment of RP4010 and afatinib clearly had a better impact on reducing intracellular Ca^2+^ oscillations than that caused by one of the single drugs alone. Similarly, we performed simulation and experimental measurement of [Ca^2+^]_cyto_ treated with combined RP4010/afatinib at other concentrations (all in µM), ranging from 0.125/0.00625 to 5.0/2.5 (Table 4 and Figure 4c). The dose-dependent curve of the period in the combination group (green curve) was above that of both the RP4010 group (blue curve) and afatinib group (red curve), thus indicating that the combined RP4010 and afatinib may have a greater impact on intracellular Ca^2+^ oscillations than any single agent. A further analysis using the CompuSyn software showed that the combination index (CI) of RP4010 and afatinib is less than 1 (Appendix A). Similar results were observed in another ESCC cell line (Appendix A).

### 2.5. Synergistical Effects of RP4010 and Afatinib on Cytotoxicity in ESCC Cells

The intracellular Ca^2+^ oscillations regulate cell proliferation and apoptosis in ESCC cells [13]. We examined the combined effects of RP4010 and afatinib on cancer-cell viability. KYSE-150 cells were treated with 1 µM RP4010, 0.5 µM afatinib or a combination for 48 h. The cells were stained with cell-permeable nuclear dye Hoechst 33342 to visualize the nuclear morphology. While the cells in the vehicle control group showed complete and intact nuclear envelope and normal morphology (Figure 5a, left panel) and 1 µM RP4010 or 0.5 µM afatinib groups showed slightly altered nuclear morphology, such as nuclear shrinkage (middle two panels), combination treatment of 1 µM RP4010/0.5 µM afatinib induced chromatin condensation, DNA fragmentation and apoptotic body formation (right panel, indicated by white arrow heads). All the nuclear characteristic morphological features indicated that these cells underwent apoptosis. Then MTT assay was employed to evaluate the cell viability, and the dose-dependent curves were analyzed in KYSE-150 cells treated with either RP4010 or afatinib individually or in combination with each other (Figure 5b). The concentrations in the RP4010 group (blue curve) were 10, 5, 2.5, 1.25 and 0.625 µM, respectively. The concentrations in afatinib group (red curve) were 5, 2.5, 1.25, 0.625 and 0.3125 µM, respectively. The combination group (green curve) contained RP4010/afatinib (2:1) at concentrations from 5 µM/2.5 µM to 0.625 µM/0.3125 µM (as in Table 2). While RP4010 or afatinib potently reduced cell viability in KYSE-150 cells in a dose-dependent manner (Figure 5b), their combination demonstrated enhanced inhibitory function on cell viability, especially in the low concentrations (Figure 5b, green curve). To further evaluate whether the enhanced function in combined treatment is through simple addition or synergy, we performed a combination index (CI) analysis by using the CompuSyn platform. The CIs at the tested concentrations were less than 1 (CI < 1), indicating that RP4010 and afatinib worked synergistically to inhibit cell proliferation in KYSE-150 cells.

## 3. Discussion

Targeting SOCE-mediated Ca^2+^ signaling in cancer cells is an emerging chemotherapy approach, and several SOCE blockers, including RP4010, are currently being evaluated in clinical trials [23]. As a second-generation TKI, afatinib received FDA approval to be used in targeted therapy for patients with EGFR mutation-positive cancers, but not for ESCC. To improve the treatment response and patient survival rate, it is a common practice to combine several treatments or several chemotherapeutic drugs. In this study, we examined whether a combination of afatinib and RP4010 could achieve better anticancer effects in ESCC cells. Using both mathematical simulation and live-cell intracellular Ca^2+^ measurement, we evaluated the inhibitory effect of afatinib and RP4010 on intracellular Ca^2+^ oscillations in KYSE-150 cells. This computer-aided mathematical model successfully predicted the frequency of intracellular Ca^2+^ oscillations in KYSE-150 cells responding to RP4010 and afatinib either individually or in combination. Both experimental and mathematical simulation data showed that a combination of afatinib and RP4010 could synergistically reduce the frequency of intracellular Ca^2+^ oscillations and helped to predict their effects on cell viability in KYSE-150 cells. This intracellular Ca^2+^ dynamic-based mathematical simulation approach could be used as a rapid and cost-effective evaluation of combined targeting therapy drugs.

The anticancer function of TKI has been mainly studied in the aspect of inhibiting downstream PI3K/Akt and MEK/ERK pathways; however, its PLC-γ axis has received less attention. During the preparation of this article, Kim et al. reported that gefitinib, a first-generation reversible TKI, inhibits EGF-stimulated intracellular Ca^2+^ oscillations in non-small-cell lung cancer cells and that restricting extracellular Ca^2+^ can consequently enhance gefitinib sensitivity [24]. As an irreversible TKI, afatinib potently inhibits signaling from all EGFR family receptor homodimers and heterodimers and the downstream events. Our experimental data showed that afatinib could reduce the frequency of intracellular Ca^2+^ oscillations in a dose-dependent manner (Figure 1c, Figure 2d and Figure 4c). The working model is that afatinib can regulate the Ca^2+^ release from ER through the EGFR–PLC–PIP2–IP3–IP3R axis and, thus, reshape the intracellular Ca^2+^ oscillations (illustration in Figure 6). Another major factor to control the intracellular Ca^2+^ oscillations is extracellular Ca^2+^ influx through the SOCE pathway, which is supported by our earlier reports [13,14]. This notion was further confirmed by this study. The data showed that RP4010 could potently inhibit the frequency of intracellular Ca^2+^ oscillations in ESCC cells in a dose-dependent manner (Figure 4c). It is worthwhile to note that SOCE-mediated Ca^2+^ influx regulates AKT and ERK1/2, two key molecules in downstream signaling pathways of EGFR, and activation of EGFR tyrosine kinase triggers the phosphorylation of STIM1 at ERK1/2 target sites to active SOCE [25]. Therefore, besides intracellular Ca^2+^, the crosstalk between EGFR and SOCE signaling pathways may have other interacting points, and these points require further investigation.

The intracellular Ca^2+^ oscillations are the event downstream of both EGFR and SOCE signaling pathways and control consequent gene transcription through Ca^2+^-dependent enzymes, including NF-κB, MAPK/ERK, CAMKII, etc. [26] Therefore, the intracellular Ca^2+^ oscillations could be used as a readout to evaluate any chemotherapeutic drug targeting EGFR or SOCE pathways. Fluorescence-based live-cell imaging can conveniently record the dynamics of intracellular Ca^2+^ measurement and can be easily up scaled as high-throughput analysis for drug screening. Another advantage using intracellular Ca^2+^ oscillations as a readout for drug evaluation is that a mathematical-modeling-based toolbox can be established to simulate [Ca^2+^]cyto. While the experimental data on multiple drugs are difficult to obtain and many of the drugs are expensive to acquire, the computer-aid mathematical simulation could greatly facilitate the mechanistic understanding of cross-talk between two pathways and provide rapid cost-effective test on hundreds of drugs and countless combinations with different ratios and concentrations. In this study, we established a mathematical model to simulate the intracellular Ca^2+^ oscillations by using a set of parameters obtained from experimental data (Figure 3 and Figure 4). The frequency of intracellular Ca^2+^ oscillations was the focus to adjust the parameters, using experimental data from a single drug, either RP4010 or afatinib. Then the mathematical model successfully predicted the synergetic action when both drugs were included into consideration, suggesting that this mathematical simulation system could provide reliable evaluation on the anticancer effects of combined TKIs and SOCE blockers.

Both the intracellular Ca^2+^ oscillations and cell-viability analysis showed that there is a synergetic effect between RP4010 and afatinib (Figure 4 and Figure 5c). This could be explained by the crosstalk between EGFR and SOCE signaling pathways. The aforementioned issue in many failed TKI clinical trials in ESCC is due to short-term adaptive responses and long-term acquired resistant of cancer cells to TKIs. Targeting both EGFR and SOCE simultaneously will provide synergetic anticancer effects to kill the cancer cells and effectively avoid their adaptive responses. Further studies are urgently needed to explore the combined TKIs and SOCE blockers for better chemotherapy, not only for esophageal cancer but also for other cancers, as well.

This study has two limitations. First, this mathematical model was able to simulate the frequency but not the shape and amplitude of Ca^2+^ oscillations. The model used here was a modified version of one developed by Sneyd et al. [6], with influence from Dupont et al. [27]. The frequency modulation of signal transduction includes NF-κB, MAPK, etc. [24], and our previous studies showed that the main downstream signaling pathway of Ca^2+^ oscillations in ESCC cells is NF-κB [14]. Therefore, we selected this relatively simple deterministic model to guide the experimental drug combination design. In the future, the model should be improved to include biochemical reactions, stochasticity, and to add spatial effects and other factors to better simulate both frequency and amplitude/shape of Ca^2+^ oscillations. Second, this study was conducted only in cultured human ESCC cells without in vivo analysis. Future studies on the combined chemotherapy effects of afatinib and RP4010 are warranted to be conducted in ESCC animal models, such as orthotopic xenografted mice or NMBA-induced rats. It will also be interesting to use this mathematical simulation system to evaluate many FDA-approved TKIs other than afatinib and SOCE blockers. Regardless of the limitations, this study demonstrated that a simulation model of the intracellular Ca^2+^ oscillations has the potential to be developed as an effective and rapid evaluation tool to be used to study whether the effects of the combination of drugs are synergistic, antagonistic or additive and to aid selection of the combinations of drugs producing the desired effects with the minimal dose. It could provide guidance for clinicians to determine optimal combination of TKIs and SOCE blockers.

## 4. Materials and Methods

### 4.1. Materials

RP4010 was obtained from Rhizen Pharmaceuticals, S.A (La Chaux-de-Fonds, Switzerland). The compound was dissolved in DMSO to make up a 10 µM stock solution. Afatinib was bought from Selleckchem company (Houston, TX, USA) and dissolved in DMSO to make a stock solution of 10 µM. Human ESCC (KYSE-30, KYSE-150, KYSE-70 and KYSE-790) and normal epithelial cell line (Het-1A) were used in this study [13].

### 4.2. Cell Culture

ESCC and Het-1A cells were cultured at 37 °C in a 5% CO_2_ incubator and maintained in a 1:1 mixture of RPMI-1640 medium and Ham’s F12 Medium (Corning, NY, USA) supplemented with 5% fetal bovine serum (VWR, Radnor, PA, USA) and 1% penicillin/streptomycin (Corning, NY, USA).

### 4.3. Intracellular Ca^2+^ Oscillations Measurement

KYSE-150 cells were seeded in a black 96-well plate with a clear bottom. After attachment, cells were treated with RP4010 and afatinib, combined or separately, at different concentrations (details are shown in Table 1), for 4 h. For RP4010, the concentration varied from 5 to 0.25 µM. The concentration of afatinib varied from 5 to 0.25 µM. For the combination of RP4010 and afatinib, the ratio of RP4010 to afatinib was 2:1. KYSE-150 cells were loaded with 5 µM Fluo-4 in 96-well imaging plates (BD Biosciences, Franklin Lakes, NJ, USA) at room temperature for 20 min. After washing, cells were kept in culture medium without phenol red. The intensity of fluorescent signals was recorded by using a Hamamatsu digital camera C11440 complemented with DMi8 inverted microscope (Leica, Wetzlar, Germany) with 20x objective (dry lens, NA 0.75). Time-lapse live-cell images were recorded every 5 s, for a total time period of 5 min, and the period between 2 peaks (2 Ca^2+^ oscillations) was measured in order to calculate the corresponding periods.

### 4.4. Western Blot

Het-1A, KYSE-30, KYSE-150, KYSE-70 and KYSE-790 cell lines were cultured in a 6-well plate and harvest for Western blot. Cells were lysed with RIPA buffer (150 mM NaCl, 50 mM Tris-Cl, 1 mM EGTA, 1% Triton X-100, 0.1% SDS and 1% sodium deoxycholate, pH 8.0) and then supplemented with proteinase inhibitor cocktail (Sigma-Aldrich, Burlington, MA, USA). After that, protein concentrations were quantified by using a BCA kit (Thermo, Waltham, MA, USA). Primary antibodies used in this study included anti-EGFR (1:1000, Thermo, USA), anti-ERBB2 (1:1000, Thermo, Waltham, MA, USA), anti-ERBB3 (1:1000), anti-ERBB4 (1:1000) and anti-β-actin (1:1000, Proteintech, Rosemont, IL, USA). Secondary antibodies included HRP-labeled goat anti-rabbit IgG (1:5000) and anti-mouse IgG (1:5000, Cell Signaling Technology, Danvers, MA, USA). ECL substrate reagent (Cytiva Amersham, Marlborough, MA, USA) was used to visualize signals on ChemiDoc (BioRad, Hercules, CA, USA).

### 4.5. Cell Viability Measurement

Cell viability was measured by using MTT assay. KYSE-150 cells were seeded in a 96-well plate at the number of 10^4^ in each well. After attachment, the cells were treated with RP4010 and afatinib, combined or separately, at different concentrations, for 24 h. For RP4010, the concentration varied from 10 to 0.625 µM. The concentration of afatinib varied from 5 to 0.3125 µM. For the combination of RP4010 and afatinib, the ratio of RP4010 to afatinib was 2:1. After 24 h, KYSE-150 cells were incubated with medium containing 10% of 3-(4,5-dimethylthiazol-2-yl)-2,5-diphenyl-tetrazolium bromide (MTT, 5 mg/mL), at 37 °C. After 4 h, formazan was dissolved in 150 µL DMSO. Absorbance was measured at 570 nm on SpectraMax i3 (Molecular Devices, San Jose, CA, USA). The survival curve was created by Graphpad Prism 5 (San Diego, CA, USA).

### 4.6. Mathematical Modeling

The mathematical model used was a modified version of one developed by Sneyd et al., with influence from Dupont et al. [6,27]. The governing dynamical equations for the Ca^2+^ signaling pathway are as follows:$$\begin{array}{cc} \frac{dc}{dt} & =J_{IPR}\left(c,p,h\right)-J_{serca}+\delta \left(J_{in}-J_{pm}\right)\\ \frac{dc_{e}}{dt} & =\gamma \left(J_{serca}-J_{IPR}\right)\\ \frac{dh}{dt} & =\left(h_{\infty }\left(c\right)-h\right)\left(\frac{1}{\tau_{h}\left(c\right)}\right)\\ \frac{dp}{dt} & =\nu -\beta_{p}p. \end{array}$$
where *c* denotes the Ca^2+^ concentration in the cytoplasm, [Ca^2+^ ]_cyto_; *Ce* is the Ca^2+^ concentration in the ER, [Ca^2+^ ]_ER_; *p* is the concentration of IP_3_, [IP_3_]; and *h* is the rate at which Ca^2+^ can activate IP_3_Rs.

The fluxes in the above equations were given as follows:$$\begin{array}{cc} J_{IP3R} & =k_{f}P_{O}\left(c_{e}-c\right)\\ J_{serca} & =\frac{V_{p}\left(c^{2}-\bar{K}c_{e}^{2}\right)}{c^{2}+k_{p}^{2}}\\ J_{in} & =\alpha_{0}+\alpha_{1}\frac{K_{e}^{4}}{K_{e}^{4}+c_{e}^{4}}\\ J_{pm} & =\frac{V_{pm}c^{2}}{K_{pm}^{2}+c^{2}}. \end{array}$$
where the *J* terms represent the Ca^2+^ fluxes across the cell and ER membranes. *J_IP3R_* modeled the flow of Ca^2+^ from the ER through the IP3R channel and depended upon both [Ca^2+^]_cyto_ and [Ca^2+^]_ER_. As *c* decreased, the difference between *c* and *Ce* increased, which resulted in an increase for *J_IP3R_*. The parameter *k_f_* was a scaling factor used to control the IP3R density. *P_O_* represented the open probability of IP_3_R and depended on the activation and inactivation rates of IP_3_R, which were both affected by the binding of Ca^2+^ and IP*3*. The SERCA pump moved Ca^2+^ from the cytoplasm into the ER and changed depending on *c* and *Ce*. As *Ce* decreased with respect to *c*, the entire term increased. In order to model Ca^2+^ flow into the cell, *J_in_* combined α_0_, which accounted for leaks into the cytoplasm through unspecified channels, and $\alpha_{1}\frac{K_{e}^{4}}{K_{e}^{4}+c_{e}^{4}}$, which represented SOCE-mediated Ca^2+^ influx. These channels open and close in response to [Ca^2+^]_ER_, *Ce*. As *Ce* increased, the denominator in $\alpha_{1}\frac{K_{e}^{4}}{K_{e}^{4}+c_{e}^{4}}$ increased, causing the entire term to decrease, implying that the cell does not need to continue to fill up ER Ca^2+^ stores. The opposite applied if *Ce* decreased, meaning that the cell needs to increase ER Ca^2+^ stores. A Hill equation was used to model *J_pm_*, the flow of Ca^2+^ through the plasma membrane pump, with *K_pm_* being the concentration of Ca^2+^, where half of the binding sites at the pump were occupied, and *V_pm_* is the maximum capacity of the plasma pump. The parameter *γ* denoted the ratio of the volume of the cytoplasm to the volume of the ER, and the parameter *δ* was dimensionless, as it was a scale factor relating the fluxes through the plasma membrane and ER membrane. To model IP3 concentration, *β_p_* represents the speed it takes p to decay to the steady state *p_s_* and *ν = β_p_ p_s_.*

The model was further expanded upon with the introduction of the functions *α* and *β*:$$\begin{array}{cc} P_{O} & =\frac{\beta }{\beta +k_{\beta }\left(\beta +\alpha \right)}\\ \alpha & =A\left(p\right)\left(1-{\bar{m}}_{\alpha }\left(c\right){\bar{h}}_{\alpha }\left(c\right)\right)\\ \beta & =B\left(p\right){\bar{m}}_{\beta }\left(c\right)h\left(c,t\right)\\ {\bar{m}}_{\alpha }\left(c\right) & ={\bar{m}}_{\beta }\left(c\right)=\frac{c^{4}}{K_{c}^{4}+c^{4}}\\ {\bar{h}}_{\alpha }\left(c\right) & =h_{\infty }\left(c\right)=\frac{K_{h}^{4}}{K_{h}^{4}+c^{4}}\\ 1-A\left(p\right) & =B\left(p\right)=\frac{p^{2}}{K_{p}^{2}+p^{2}}\\ \tau_{h} & =\tau_{max}\frac{K_{\tau }^{4}}{K_{\tau }^{4}+c^{4}}. \end{array}$$

The function *α(c,p)* modeled the rate at which [Ca^2+^ ] inactivating IP3Rs and *β(c,p,t)* modeled the rate at which [Ca^2+^ ] activates IP_3_Rs.

### 4.7. Addition of TKI and SOCE Blocker in the Computational Model

A new feature in this computational model was the incorporation of the effects of one drug or of the combination of two different drugs acting on different Ca^2+^ resources. TKI (afatinib) and SOCE blocker (RP4010) were incorporated into the model through *D1* and *D2*, which represent the overall drug effect on each of the channels:$$\begin{array}{cc} J_{in} & =\alpha_{0}+D_{2}\alpha_{1}\frac{K_{e}^{4}}{K_{e}^{4}+c_{e}^{4}}\\ \frac{dp}{dt} & =D_{1}\nu -\beta_{p}p, \end{array}$$

As a SOCE channel inhibitor, RP4010 decreased [Ca^2+^]_cyto_ and was expected to reduce the frequency of the oscillations. As a TKI, afatinib inhibits EGFR tyrosine kinase activity and thus affects the steady state of IP3. It was expected that it negatively affected the ER Ca^2+^ release through the IP3R channel and, again, reducing [Ca^2+^]_cyto_. In order to describe the relationship between the effectiveness of the drug to the dose of drug given to the cell, we used the following equations:$$\begin{array}{cc} D_{1} & =1-f_{1}\left(S_{1}\right)\\ D_{2} & =1-\left(f_{2}\left(S_{2}\right)-f_{3}\left(S_{1}\right)\right) \end{array}$$
where the functions ***f_i_*** were described through the Michaelis–Menten kinetics:$$f_{i}\left(S\right)=c_{i}\;S/\left(k_{i}+S\right),\;i=1,2,3)$$
where ***S_1_*** and ***S_2_*** represent the doses of afatinib and RP4010, respectively.

All parameters used in the mathematical model were summarized in Table 3.

### 4.8. Validation of the Mathematical Model

We fitted the parameters of the mathematical model and validated it by using experimental data for different concentrations of drugs individually or in combination with a ratio of RP4010:afatinib = 2:1. The observed experimental data showed that [Ca^2+^] oscillates in ESCC cells had a period of 32.8 s. We used this value to adjust the model parameters *k_f_*, α1 and τmax, accordingly, to fit. In non-tumorous esophageal epithelial cells, which contain much less SOCE and ER Ca^2+^ release, we divided the values of k f by 5 and α1 by 3 to reflect such differences. The working assumption was that increasing concentrations of TKI or SOCE blockers would gradually decrease the level of intracellular Ca^2+^ oscillations in tumor cells until to the level in Het-1A cells. Finally, to find the coefficients ci and ki in the Michaelis–Menten equations, we matched the experimental data regarding drug doses and their corresponding periods.

To carry out the numerical simulations, the MATLAB’s ode23s built-in function, based on the explicit Runge–Kutta (2,3) pair of Bogacki and Shampine [28], was used to solve the systems of differential equations. The MATLAB’s lsqcurvefit built-in function was implemented by using the Levenberg–Marquardt algorithm [29] that solves a minimization problem involving a least-squares data-fitting term for fitting the model to the experimental data presented in Table 1.

### 4.9. Statistical Analysis

In this study, at least 50 cells in each concentration group were included to calculate the period (peak–peak duration) of Ca^2+^ oscillations. Cancer cells present different patterns of Ca^2+^ oscillations during cell-cycle progression, and we focused on the G(1)/S phase, a determining phase for cell proliferation that contains majority cells with Ca^2+^ oscillations. To simplify the model, we trimmed the dataset by removing any data outside 1.5 times the interquartile range. After trimming, the mean values of the period calculated from the rest of the dataset were used for the mathematical simulation, as presented in Table 1 and Table 2.

The combination index (CI) was calculated from the CI equation algorithms by using CompuSyn software (CompuSyn, Inc., Paramus, NJ, USA). Dose–effect Curve and combination index (CI) versus fraction affected (fa) plots were analyzed and exported from CompuSyn [30].

## Figures and Tables

**Figure 1 ijms-23-01763-f001:**
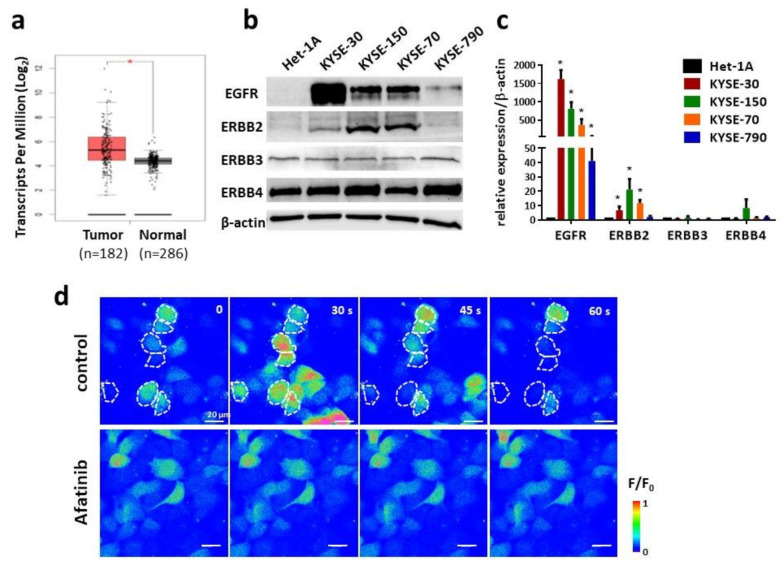
Upregulated expression of EGFR and ERBB2 and inhibited Ca^2+^ oscillations by afatinib in ESCC cells. (**a**) RNA-Seq data (TCGA) revealed significant higher expression of EGFR in tumor vs. normal tissues removed from patients. Average expression level (TPM) for tumor and non-tumor is 38.4 and 20.4, respectively. Tumor (*n* = 182), including ESCC (*n* = 94), EAC (*n* = 87) and cystic, mucinous and serous neoplasms (*n* = 1). Normal (*n* = 286), including TCGA adjacent normal (*n* = 13) and GTEx normal esophageal tissue (*n* = 273); * *p*-value < 0.0001. (**b**) Western Blot results of EGFR family members (EGFR, ERBB2, ERBB3 and ERBB4) in ESCC and Het-1A cells. β-actin was used as loading control. (**c**) Quantification of Western blot result of EGFR family members in ESCC and Het-1A cells; * *p*-value < 0.05. (**d**) Time-lapse live-cell fluorescent imaging of intracellular Ca^2+^ in KYSE-150 cells. Cells were treated with either vehicle control or 5 μM afatinib for 4 h. The intracellular Ca^2+^ concentrations were presented by a heap map with fluorescent intensity changes (F/F0). Some representative cells with obvious Ca^2+^ oscillations were indicated by dashed white circle. Scale bar, 20 μm.

**Figure 2 ijms-23-01763-f002:**
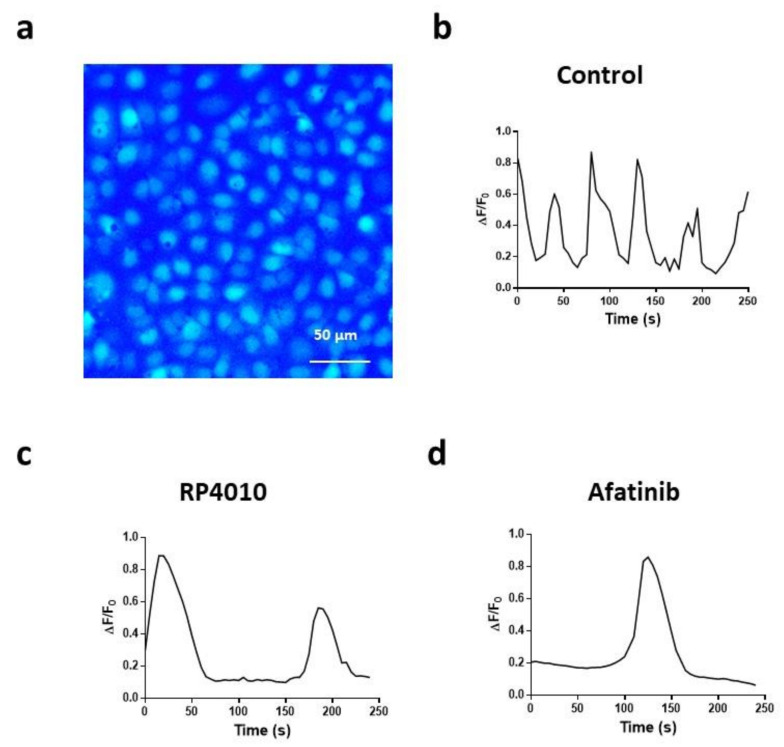
(**a**) Decreased intracellular Ca^2+^ oscillations in KYSE-150 cells treated with RP4010 or afatinib. (**b**–**d**) Fluorescent image of KYSE-150 cells loaded with Fluo-4 AM. Scale bar, 50 μm. (**b**) Representative traces of Ca^2+^ oscillations in a single KYSE-150 cell treated with vehicle control, (**c**) RP4010 or (**d**) afatinib.

**Figure 3 ijms-23-01763-f003:**
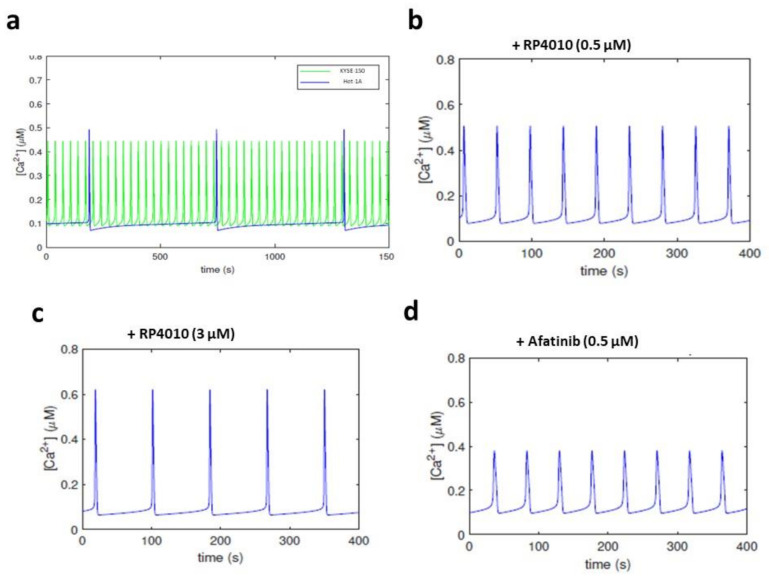
Ca^2+^ oscillations simulated by mathematical modeling. (**a**) Simulated Ca^2+^ oscillations in KYSE-150 (green line) and Het-1A cells (blue line). (**b**,**c**) Simulated Ca2+ oscillations in KYSE-150 cells treated with RP4010 at the concentration of 0.5 µM (**b**) or 3.0 µM (**c**). (**d**) Simulated Ca^2+^ dynamics in KYSE-150 cells treated with afatinib at the concentration of 0.5 µM.

**Figure 4 ijms-23-01763-f004:**
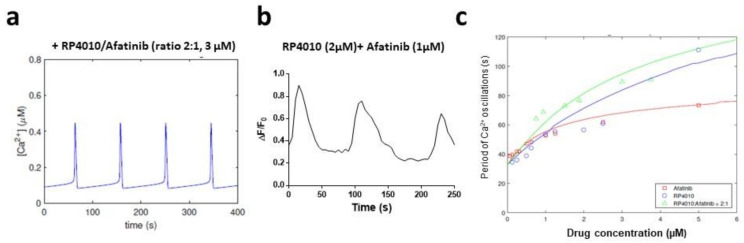
Effect of combined treatment of RP4010 and afatinib on intracellular Ca^2+^ oscillations. (**a**) Mathematical simulation of Ca^2+^ dynamics in KYSE-150 cells treated with combined RP4010 and afatinib (2:1 ration at 3.0 µM). (**b**) A representative trace of intracellular Ca^2+^ concentration measured by time-lapse live-cell imaging in KYSE150 cells treated with combined 2 µM RP4010 and 1 µM afatinib. (**c**) Dose-dependent curves of simulated periods of intracellular Ca^2+^ oscillations in KYSE-150 cells treated with either RP4010 (blue), afatinib (red) or combination (green). Individual circle, square or triangle is from experimental data.

**Figure 5 ijms-23-01763-f005:**
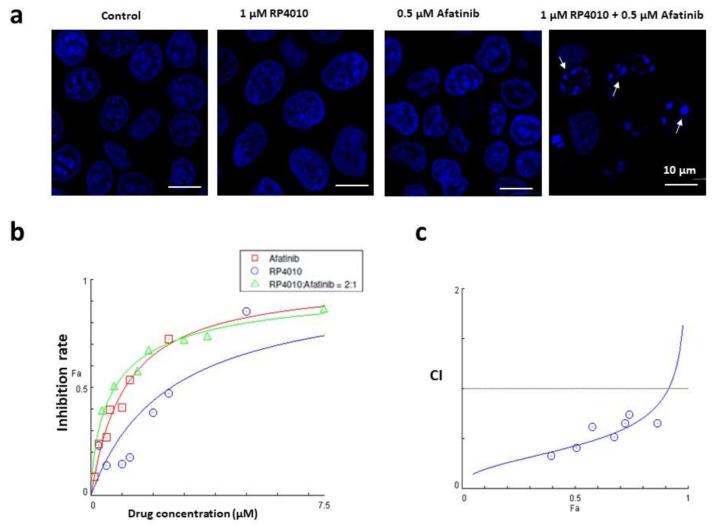
Synergetic effect of combined RP4010 and afatinib on cell viability in KYSE-150 cells. (**a**) Nuclei staining by Hoechst 33342 in KYSE-150 cells treated with vehicle control, 1 µM RP4010, 0.5 µM afatinib or 1 µM RP4010 + 0.5 µM afatinib. Apoptotic cells were indicated by nuclear DNA fragmentation and condensation (white arrow heads). Scale bar, 10 μm. (**b**) Dose-dependent inhibition curves of KYSE-150 cells treated by RP4010 (blue circles), afatinib (red squares) or RP4010 plus afatinib (ratio 2:1, green triangles). Cell inhibition rates were measured by MTT assay. Details in the confined box were amplified as the upper curve. (**c**) Combination index (CI) plot for combined effect of RP4010 and afatinib. Each blue circle indicated a data point with combined RP4010 and afatinib. Simulation curve is shown in blue line. Average effect values (Fa) are shown on *x*-axis.

**Figure 6 ijms-23-01763-f006:**
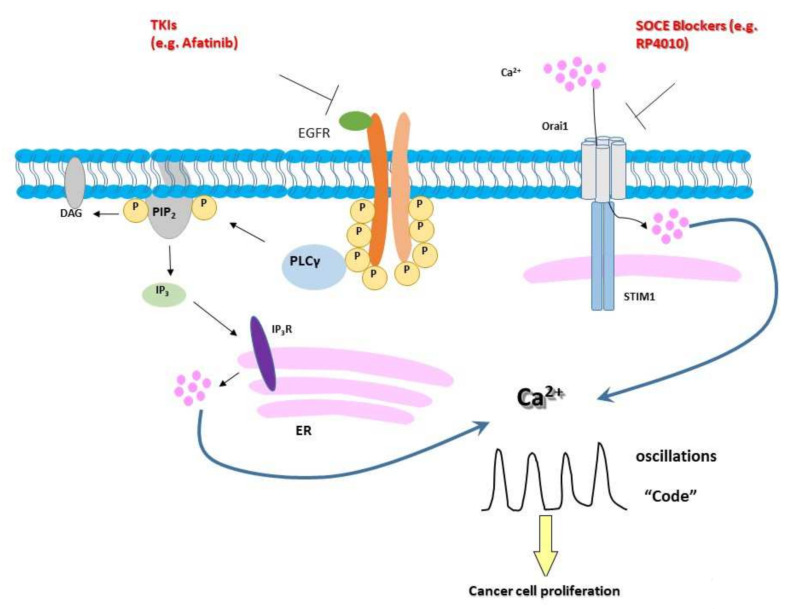
Illustration of intracellular Ca^2+^ oscillations as the intersection between EGFR and SOCE signaling pathways. [Ca^2+^]cyto are dependent on EGFR/PLCγ/IP_3_/IP_3_R-mediated ER Ca^2+^ release and Orai1-mediated SOCE. Intracellular Ca^2+^ oscillations are code, especially the frequency, to control cell proliferation in ESCC cells. Inhibiting either EGFR by Afatinib or Orai1 by RP4010 could reduce the frequency of intracellular Ca^2+^ oscillations and inhibit cancer-cell proliferations.

**Table 1 ijms-23-01763-t001:** Dose-dependent drug effects on intracellular Ca^2+^ oscillations (experimental data).

Drug Concentration (µM)	Period (s) Mean ± SEM	
	RP4010	Afatinib
10	122 ± 6.6	/
5	111 ± 6.2	73.4 ± 4.3
2.5	60.8 ± 4.0	61.9 ± 3.5
2	56.5 ± 2.7	/
1.25	55.4 ± 3.5	54.2 ± 2.5
1	53.5 ± 2.9	52.7 ± 2
0.625	44.1 ± 3.3	48.2 ± 2.4
0.5	38.8 ± 1.3	47.1 ± 1.5
0.3125	/	41.9 ± 2.6
0.25	35.9 ± 1.3	41.8 ± 1.4
0.125	34.4 ± 2	39.8 ± 1.4
0.0625	/	38.6 ± 2.2
0.0	32.8 ± 1.2	

**Table 2 ijms-23-01763-t002:** Parameter values for the mathematical model.

Parameter	Description	Value	Units
*δ*	Used to adjust ratio of [Ca^2+^] across plasma membrane to ER membrane	1.5	n/a
*Kτ*	The concentration of Ca^2+^ in response to β	0.1	µM
*Kc*	Half-maximal [Ca^2+^] for IP3R	0.2	µM
*k f*	Scaling factor that controls [Ca^2+^] release through IP3R; IP3R density and channel activity	3.9	s^−1^
*Vserca*	Maximum capacity of SERCA pump	0.9	µMs^−1^
*γ*	Ratio of cytoplasmic volume to ER volume	5.5	n/a
*Vpm*	Maximum capacity of plasma membrane pump	0.11	µMs^−1^
*α0*	Flow of calcium into the cell through an unspecified leak	0.0027	µMs^−1^
*KP*	Half-maximal [IP3] for IP3R	0.2	µM
*τmax*	Controls rate that β responds to [Ca^2+^] changes	1420	s^−1^
*βp*	Rate of decay of p to its steady state	0.027	s^−1^
*Kh*	The concentration of Ca^2+^ activated IP3R	0.08	µM
*K¯*	Used to adjust the Ca^2+^ concentration in ER	1.9 × 10^−5^	n/a
*Ke*	Half-maximal [Ca^2+^] ER for SOC channels	8	µM
*Kpm*	Half-maximal [Ca^2+^] for plasma membrane pump	0.3	µM
*α1*	Rate constant for SOC channels	0.385	µMs^−1^

**Table 3 ijms-23-01763-t003:** Summary of sensitivity analysis results.

Dose (µM)	Parameter	Baseline	*p*-Value	PRCC Value
Aft: 0.5 RP: 1.0	kf	3.9	0.643	0.034
	α1	0.385	1.44 × 10^−50^	−0.831
	c1	0.51	0.272	0.079
	c2	1	1.74 × 10^−46^	0.812
	c3	0.76	4.6 × 10^−16^	−0.541
	k1	0.34	0.061	−0.1353
	k2	0.6	4.01 × 10^−12^	−0.4724
	k3	0.8	8.45 × 10^−8^	0.3740

**Table 4 ijms-23-01763-t004:** Effects on intracellular Ca^2+^ oscillations by combined RP4010 and afatinib (experimental data).

Concentration of Combined Drugs (µM)		Period (s)
RP4010	Afatinib	
5	2.5	92.3 ± 4.6
2.5	1.25	90.8 ± 4.4
2	1	89.4 ± 6.4
1.5	0.75	76.8 ± 4.6
1.25	0.625	73.1 ± 3.8
1	0.5	68.7 ± 3.9
0.625	0.3125	64.2 ± 3.7
0.25	0.125	61.1 ± 3.1
0.125	0.0625	59.6 ± 2.5

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
