# Peer review of "Developing a Mathematical Model of Intracellular Calcium Dynamics for Evaluating Combined Anticancer Effects of Afatinib and RP4010 in Esophageal Cancer"

_ijms, 2022, doi:10.3390/ijms23031763_

Round 1

Reviewer 1 Report

Specific comments to the authors

The authors Yan Chang et al. of the submitted manuscript „Experimental and mathematical models of intracellular calcium dynamics for evaluating combined anticancer effects of Afatinib and RP4010 in esophageal cancer” studied the influence of calcium dynamics on anticancer effects of Afatinib and RP4010. In summary, based on their study design using a combination of mathematical and experimental approach the authors could demonstrate that the developed mathematical simulation of Ca2+ oscillations correlate with the experimental data responding to afatinib or RP4010, separately or in combination. Therefore, the authors postulated (i) that combination of afatinib 24 and RP4010 represent a synergistic anticancer effect and (ii) that the intracellular Ca2+ dynamic-based mathematical simulation approach is suitable for a rapid and cost-effective evaluation of combined targeting therapy drugs.

Overall, the manuscript give interesting information how mathematical simulation could be used for combinatory drug-therapy development in future, whereby additional theoretical and experimental control analysis to verify or to falsify their interesting mathematical simulation of in-vitro-findings. Furthermore, the anti-cancer combinatory effects of combined RP4010 and afatinib treatment on esophageal squamous cancer cells are interesting too, whereby the insight mechanism keep unclear. The manuscript (including presentation) is comprehensible and convincing. The methods are mostly well described. Although the results and discussion are clear presented, the authors (see specific comments) must perform some major changes to improve the manuscript. In conclusion, the presented data are interesting. After incorporating the mentioned specific comments (see below) the manuscript has the potency to be accepted.

Specific comments

Title: The title does not reflect the content of the submitted manuscript, since the key message is the development of a mathematical model for predicting the effect of two inhibitory drugs, alone and in combination. Please change adequately.

Abstract: Please specify the “results” in the sentence “The results showed that combination of afatinib and RP4010 presented synergistic anticancer effect.”. Additionally, please specify the used human cancer cell lines, too.

Material & Methods: Please add information of the in-silico-analysis as mentioned below in the comments to figure 1.

Results:

# Table 2: By reading the manuscript, it is not clear, where the parameters of the used mathematical modell are derived from. Please clear it out in more detail.

# Figure 1: Interestingly, the authors performed in-silico-analysis, which should be described in more detail in the material&method section. Please quantify the Western blottings for further statistical analysis. How is the EGFR mutation status of the applied esophageal squamous cancer cell lines? It is possible to have more broader time range for the effects of Afatinib?

# Figure 2: Regarding the figure 2, it should be of interest to investigate the effect of different extracellular Ca+-concentration, too. 

# Figure 4: Please argue why a “molar ratio 2:1 for RP4010 and afatinib” was selected.

# Figure 5: Additive or synergistic effects must be “classically” investigated by comprehensive isobologram analysis. More experiments are needed to specify the anti-proliferative and the pro-apoptotic effects of the combination of effect of combined RP4010 and afatinib.

Discussion: Overall, the discussion is largely repetitive, narrative and sometimes speculative. The authors should focus on and highlight more the own very interesting of mathematical modelling in the chosen in-vitro human cancer model. In detail, the sentence “Both experimental and mathematical simulation data showed that combination of afatinib and RP4010 could synergistically reduce the frequency of intracellular Ca2+ oscillations and thus cell viability in KYSE-150 cells.” is not supported by the experiments: the mathematical simulation could predict the Ca2+-oscillations, but not the viability. This should be cleared by the authors. Furthermore, the sentence “The intracellular Ca2+ oscillations are the event downstream of both EGFR and SOCE signaling pathways and serve as “code” to control consequent gene transcription and cell proliferation, cell migration/invasion.” sounds largely dubious and not scientific. Therefore, please discuss possibly Ca2+ dependent downstream mechanism leading to decreased proliferation or increased apoptosis. Finally, the authors should discuss the limitations of the presented study in detail (no in-vivo or in-situ analysis). How could the interesting findings transferred from a theoretical to a practical view (like clinical setting (drug-development, drug-combination))? Please discuss in short.

Author Response

We appreciate this reviewer for your constructive and insightful critiques.

Overall, the manuscript give interesting information how mathematical simulation could be used for combinatory drug-therapy development in future, whereby additional theoretical and experimental control analysis to verify or to falsify their interesting mathematical simulation of in-vitro-findings. Furthermore, the anti-cancer combinatory effects of combined RP4010 and afatinib treatment on esophageal squamous cancer cells are interesting too, whereby the insight mechanism keep unclear. The manuscript (including presentation) is comprehensible and convincing. The methods are mostly well described. Although the results and discussion are clear presented, the authors (see specific comments) must perform some major changes to improve the manuscript. In conclusion, the presented data are interesting. After incorporating the mentioned specific comments (see below) the manuscript has the potency to be accepted.

Response: We appreciate this reviewer’s encouraging comments.

Specific comments

Title: The title does not reflect the content of the submitted manuscript, since the key message is the development of a mathematical model for predicting the effect of two inhibitory drugs, alone and in combination. Please change adequately.

Response: Thank you for the suggestion. We now change the title to “Developing a mathematical model of intracellular calcium dynamics for evaluating combined anticancer effects of Afatinib and RP4010 in esophageal cancer”.

Abstract: Please specify the “results” in the sentence “The results showed that combination of afatinib and RP4010 presented synergistic anticancer effect.” Additionally, please specify the used human cancer cell lines, too.

Response: Following your suggestion, we added cancer cell line KYSE-150 and provided specificity to the result. The changes are highlighted in Red as following.

“…we evaluated the inhibitory effect of afatinib and RP4010 on intracellular Ca2+ oscillations in KYSE-150, a human esophageal squamous cell carcinoma cell line, using both experimental and mathematical simulations. Our mathematical simulation of Ca2+ oscillations could fit well with experimental data responding to afatinib or RP4010, separately or in combination. Guided by simulation, we were able to identify a proper ratio of afatinib and RP4010 for combined treatment and such combination presented synergistic anticancer effect evidence by experimental measurement of intracellular Ca2+ and cell proliferation.”

Results:

# Table 2: By reading the manuscript, it is not clear, where the parameters of the used mathematical model are derived from. Please clear it out in more detail.

Response: We thank the Reviewer for this comment. The parameter values in Table 2 were taken from the model presented by Sneyd et. al. [6] with the exceptions of kf, α1. These were the parameters changed to adjust the model to fit the type of cells used in the experiments. This has now been made more clear in the revised version of the manuscript (Line 163-166).

# Figure 1: Interestingly, the authors performed in-silico-analysis, which should be described in more detail in the material&method section. Please quantify the Western blottings for further statistical analysis.

Response: Thank you for the suggestion. The Western blot have been done with 3 biological replicates and total 5 samples. Densitometry calculation using ImageJ were used to perform statistical analysis (please see new Figure 1c). 

How is the EGFR mutation status of the applied esophageal squamous cancer cell lines?

Response: TCGA data and previous studies have showed that human esophageal squamous cell carcinoma cells presented much less mutation (distinct from lung cancer and breast cancer), rather, the amplification copy number are much higher. Our recent published study not only confirmed the higher expression of wild type EGFR but also identified a novel enhancer RNA for EGFR in KYSE-150 and KYSE-30 cells by Assay for Transposase-Accessible Chromatin using sequencing (ATAC-seq) (Choi et al, 2021). Afatinib is one of the second generation TKIs and has an irreversible inhibition on mutated or wild type but increased expression of EGFR. Thus, the EGFR mutation status of the proposed esophageal cancer cell lines will have no impact on the inhibitory effects of afatinib. In the future, the mutation status of different type of cancers, which alters TKI’s inhibitory property, will be included into consideration. This is also our long-term goal to develop a simple and robust simulation model for drug prescreening and evaluation.

Ref:

Choi S, Sathe A, Mathe E, Xing C*, Pan Z*. Identification of a new Enhancer RNA for EGFR in hyper-accessible regions in esophageal squamous cell carcinoma cells by analysis of chromatin accessibility landscapes. Frontiers in Oncology, 2021. Oct 15. doi: 10.3389/fonc.2021.724687.

It is possible to have more broader time range for the effects of Afatinib?

Response: Since longer treatment time may result adaptive drug resistance to complicate our analysis, we purposely tested the effects of afatinib up to 96 hours in current study.

# Figure 2: Regarding the figure 2, it should be of interest to investigate the effect of different extracellular Ca+-concentration, too. 

Response: The effect of extracellular Ca2+ concentrations on intracellular Ca2+ oscillations were previously studied (ref 13). Addition of Ca2+ chelator EGTA (0.5mM) silenced these oscillations. Removal of extracellular Ca2+ concentrations may impact multiple targets, including store-operated Ca2+ entry channel (SOCE), extracellular calcium-sensing receptor (CaSR), Transient Receptor Potential Channels (TRP), which in turn will change intracellular Ca2+ signals. Since our study was focused on two druggable targets, i.e. SOCE and EGFR, we used more specific inhibitors to study their effects on intracellular Ca2+ oscillations rather than manipulation of extracellular Ca2+ concentrations.

# Figure 4: Please argue why a “molar ratio 2:1 for RP4010 and afatinib” was selected.

Response:  Our hypothesis is that RP4010 and Afatinib can work synergistically to inhibit esophageal cancer cell proliferation. The basic test method involved identification of IC50 of each drug and then test the combined effect of both drugs at the determined molar ratio (at IC50). The simulation data of inhibitory effects on intracellular Ca2+ oscillations showed that the IC50 of RP4010 and afatinib were 1.4 μM and 0.7μM, respectively. Therefore, a molar ration 2:1 for RP4010 and afatinib were selected.

# Figure 5: Additive or synergistic effects must be “classically” investigated by comprehensive isobologram analysis. More experiments are needed to specify the anti-proliferative and the pro-apoptotic effects of the combination of effect of combined RP4010 and afatinib.

Response: We appreciated the professional comments. Comprehensive isobologram analysis is a very scientific method that is widely used in drug evaluations [Huang, et al]. Huang et al. also pointed out that “Combination index (CI) is used to determine the degree of drug interaction, and its formula is the sum of the ratio of the dose of each drug in the compound to the dose when used alone when the combination and compound produce 50% efficacy. The principle and application are the same as that in isobologram.” [1] In figure 5C, we used Compusyn to generate CI and evaluate the drug interaction. In Compusyn software generated results, combination index (CI) theorem of Chou-Talalay offers quantitative definition for additive effect (CI = 1), synergism (CI < 1), and antagonism (CI > 1) in drug combinations [Chou, et al]. From the Figure 5c, all the drug combined points are below 1 which shows the synergetic effects. 

Ref:

  1. Huang, R.Y.; Pei, L.; Liu, Q.; Chen, S.; Dou, H.; Shu, G.; Yuan, Z.X.; Lin, J.; Peng, G.; Zhang, W.; et al. Isobologram Analysis: A Comprehensive Review of Methodology and Current Research. Frontiers in pharmacology 2019, 10, 1222, doi:10.3389/fphar.2019.01222.
  2. Chou, T.C. Drug combination studies and their synergy quantification using the Chou-Talalay method. Cancer research 2010, 70, 440-446, doi:10.1158/0008-5472.CAN-09-1947.

Discussion: Overall, the discussion is largely repetitive, narrative and sometimes speculative. The authors should focus on and highlight more the own very interesting of mathematical modelling in the chosen in-vitro human cancer model. In detail, the sentence “Both experimental and mathematical simulation data showed that combination of afatinib and RP4010 could synergistically reduce the frequency of intracellular Ca2+ oscillations and thus cell viability in KYSE-150 cells.” is not supported by the experiments: the mathematical simulation could predict the Ca2+-oscillations, but not the viability. This should be cleared by the authors. Furthermore, the sentence “The intracellular Ca2+ oscillations are the event downstream of both EGFR and SOCE signaling pathways and serve as “code” to control consequent gene transcription and cell proliferation, cell migration/invasion.” sounds largely dubious and not scientific. Therefore, please discuss possibly Ca2+ dependent downstream mechanism leading to decreased proliferation or increased apoptosis. Finally, the authors should discuss the limitations of the presented study in detail (no in-vivo or in-situ analysis). How could the interesting findings transferred from a theoretical to a practical view (like clinical setting (drug-development, drug-combination))? Please discuss in short.

Response: We appreciate the valuable critiques. We are sorry for the redundant paragraphs in Discussion due to errors in format converting. We deleted redundant paragraphs, rephrased the sentence to reflect the study exactly, and added discussion on the limitation of this study as well as how to transfer this interesting finding from theoretical to clinical use (Line 434-453). Changes are marked in Red font.

“Both experimental and mathematical simulation data showed that combination of afatinib and RP4010 could synergistically reduce the frequency of intracellular Ca2+ oscillations, which helped to predict their effects on cell viability in KYSE-150 cells.“ (Line 304-307)

“The intracellular Ca2+ oscillations are the event downstream of both EGFR and SOCE signaling pathways and control consequent gene transcription through Ca2+-dependent enzymes, including AKT and ERK1/2.” (Line 336-339)

Reviewer 2 Report

The manuscript describes a combined experimental and computational study into intracellular calcium oscillations in KYSE150 cells with a focus on esophageal cancer. The authors' previous study suggested that intracellular calcium oscillations are an important  code for activation of downstream signaling pathways in esophageal squamous carcinoma  cells. In the present study, the authors investigate in more detail the action of two inhibitors of the calcium signalling toolkit: RP4010, which blocks store-operated-calcium entry (SOCE), and afatinib, a tyrosine kinase inhibitor, which interacts with  inositol-1,4,5-trisphosphate (IP3) production. A main result of this study is that the two inhibitors can work synergistically and in isolation in reducing the frequency of calcium oscillations, which has been shown to have an anti-cancer effect. I completely agree with the authors that targeting the calcium signalling toolkit to combat cancer is a worthwhile and promising approach. I also believe that combining experimental and computational studies provides a powerful and insightful approach to better understand biomedical processes and to build a framework for efficient testing of hypotheses and making predictions. However, there are a few points that I would like the authors to address before the manuscript can be considered for publication. 

- The main focus of the manuscript is calcium oscillations, but there is very little background on modelling calcium oscillations. Given that one of the main results of the manuscript is the mathematical model, this needs to be rectified. Although oscillations of the intracellular calcium concentration have been modelled for decades, controversies still abound. In particular, whether it is appropriate to describe these oscillations, which have been shown to be stochastic in many cell types, by a deterministic set of ODEs as in the present study. I recommend that the authors justify their model more. A mere "The mathematical model used was a modified version of one developed by Sneyd et al. with influence from Dupont et al. [6,25]." is hardly sufficient. Here is a list of references that should be included and discussed.
- Dupont, G., & Sneyd, J. (2017). Recent developments in models of calcium signalling. Current Opinion in Systems Biology, 3, 15–22. http://doi.org/10.1016/j.coisb.2017.03.002
- Voorsluijs, V., Dawson, S. P., De Decker, Y., & Dupont, G. (2019). Deterministic Limit of Intracellular Calcium Spikes. Physical Review Letters, 122(8), 088101. http://doi.org/10.1103/PhysRevLett.122.088101
- Dupont, G., Combettes, L., Bird, G. S., & Putney, J. W. (2011). Calcium oscillations. Cold Spring Harbor Perspectives in Biology, 3(3). https://doi.org/10.1101/cshperspect.a004226
- Wacquier, B., Voorsluijs, V., Combettes, L., & Dupont, G. (2019). Coding and decoding of oscillatory Ca2+ signals. Seminars in Cell & Developmental Biology, 94, 11–19.
- Rüdiger, S. (2014). Stochastic models of intracellular calcium signals. Physics Reports-Review Section of Physics Letters, 534(2), 39–87. http://doi.org/10.1016/j.physrep.2013.09.002

- In a similar vein, there should be references to earlier computational models of SOCE, such as
- Gil, D., Guse, A. H., & Dupont, G. (2021). Three-dimensional model of sub-plasmalemmal Ca2+ microdomains evoked by the interplay between ORAI1 and InsP3 receptors. Frontiers in Immunology, 12, 659790.

- It is good to see a sensitivity analysis, but it should be discussed more. Also, given that the ODE model is quite cheap to simulate, a more comprehensive analysis would be desirable. The authors should also comment on why they do not change more than one parameter at the time.

- The authors mention that they obtained some parameter values from fitting their model to experimental data. More details are needed. The only reference to a MATLAB function is not enough to understand the fitting procedure and the quality of the fit.

- A ratio of cytoplasmic volume to ER volume of 5.5 seems rather small. Could the authors provide a justification for this and put in into context with measured volume ratios?

- Figure 4 compares simulation results with experimental data. While the frequency of the oscillations is comparable, the shape of the oscillations does not match at all. While one could argue that the manuscript is primarily concerned with the frequency of the oscillations, the wave form matters (see e.g. Dupont, G., & Combettes, L. (2016). Fine tuning of cytosolic Ca2+ oscillations. F1000Research, 5. http://doi.org/10.12688/f1000research.8438.1). The reasons for the discrepancy between model and data needs to be discussed.

Author Response

We appreciate this reviewer for your recognition of the significance of this work. Their insightful and constrictive comments and suggestions have helped us to improve this manuscript. Our point-by-point responses to the comments and suggestions are listed as below. 

The manuscript describes a combined experimental and computational study into intracellular calcium oscillations in KYSE150 cells with a focus on esophageal cancer. The authors' previous study suggested that intracellular calcium oscillations are an important code for activation of downstream signaling pathways in esophageal squamous carcinoma  cells. In the present study, the authors investigate in more detail the action of two inhibitors of the calcium signalling toolkit: RP4010, which blocks store-operated-calcium entry (SOCE), and afatinib, a tyrosine kinase inhibitor, which interacts with inositol-1,4,5-trisphosphate (IP3) production. A main result of this study is that the two inhibitors can work synergistically and in isolation in reducing the frequency of calcium oscillations, which has been shown to have an anti-cancer effect. I completely agree with the authors that targeting the calcium signalling toolkit to combat cancer is a worthwhile and promising approach. I also believe that combining experimental and computational studies provides a powerful and insightful approach to better understand biomedical processes and to build a framework for efficient testing of hypotheses and making predictions. However, there are a few points that I would like the authors to address before the manuscript can be considered for publication. 

Response: We are so delighted to know that this reviewer agrees that targeting the calcium signaling toolkit to computational studies provides a powerful and insightful approach to better understand pathophysiology.

- The main focus of the manuscript is calcium oscillations, but there is very little background on modelling calcium oscillations. Given that one of the main results of the manuscript is the mathematical model, this needs to be rectified. Although oscillations of the intracellular calcium concentration have been modelled for decades, controversies still abound. In particular, whether it is appropriate to describe these oscillations, which have been shown to be stochastic in many cell types, by a deterministic set of ODEs as in the present study. I recommend that the authors justify their model more. A mere "The mathematical model used was a modified version of one developed by Sneyd et al. with influence from Dupont et al. [6,25]." is hardly sufficient. Here is a list of references that should be included and discussed.
- Dupont, G., & Sneyd, J. (2017). Recent developments in models of calcium signalling. Current Opinion in Systems Biology, 3, 15–22. http://doi.org/10.1016/j.coisb.2017.03.002
- Voorsluijs, V., Dawson, S. P., De Decker, Y., & Dupont, G. (2019). Deterministic Limit of Intracellular Calcium Spikes. Physical Review Letters, 122(8), 088101. http://doi.org/10.1103/PhysRevLett.122.088101
- Dupont, G., Combettes, L., Bird, G. S., & Putney, J. W. (2011). Calcium oscillations. Cold Spring Harbor Perspectives in Biology, 3(3). https://doi.org/10.1101/cshperspect.a004226
- Wacquier, B., Voorsluijs, V., Combettes, L., & Dupont, G. (2019). Coding and decoding of oscillatory Ca2+ signals. Seminars in Cell & Developmental Biology, 94, 11–19.
- Rüdiger, S. (2014). Stochastic models of intracellular calcium signals. Physics Reports-Review Section of Physics Letters, 534(2), 39–87. http://doi.org/10.1016/j.physrep.2013.09.002

- In a similar vein, there should be references to earlier computational models of SOCE, such as
- Gil, D., Guse, A. H., & Dupont, G. (2021). Three-dimensional model of sub-plasmalemmal Ca2+ microdomains evoked by the interplay between ORAI1 and InsP3 receptors. Frontiers in Immunology, 12, 659790.

Response: We appreciate this reviewer’s encouraging comments.

The mathematical model used was a modified version of one developed by Sneyd et al. with influence from Dupont et al. [6, iv, vi]. The main objective of this relatively simple deterministic model is to guide the experimental work. The model can be written as biochemical reactions and stochasticity added, spatial effects and other factors included, but this will make the model also more complex. As pointed in Rudiger et al [ii] and references therein mention that the number of molecules  involved in intracellular reactions, specifically those involving Ca2+ dynamics is small, so there are large variations, the events are stochastic and the use of the averaged differential equation model is not adequate. The Stochastic Simulation Method also known as Gillespie’s Algorithm [i] is widely used for stochastic simulations of biochemical reactions, and in our future research we will add stochasticity. But it also worth mentioning that Voorsluijs et al. demonstrated that under certain conditions the deterministic and stochastic approaches can be reconciled [iii]. See also the review by Dupont and Sneyd [iv].  Also for more information see the papers by Wacquier et al.[v], who found that numerous modulating factors, such as the dynamics of IP3 or the involvement of other Ca2+ pools affect the calcium oscillations, and by Dupont et al. [vi] who present a table classifying the main types of computational models for intracellular Ca2+ oscillations. They also show how the use of computational models of Ca2+ oscillations can help understand the mechanisms of generation of oscillations and review recent findings indicating an important role for store operated Ca entry in Ca2+ oscillations.

Ref.

  1. D. Gillespie. A general method for numerically simulating the stochastic time evolution of coupled chemical reactions. Journal of Computational Physics, 22 (4) (1976), pp. 403-434

  1. Rüdiger, Sten. “Stochastic Models of Intracellular Calcium Signals.” Physics Reports 534, no. 2 (January 2014): 39–87. https://doi.org/10.1016/j.physrep.2013.09.002.

  • Voorsluijs, V., S. Ponce Dawson, Y. De Decker, and G. Dupont. “Deterministic Limit of Intracellular Calcium Spikes.” Physical Review Letters 122, no. 8 (February 26, 2019): 088101. https://doi.org/10.1103/PhysRevLett.122.088101.

  1. Dupont, Geneviève, and James Sneyd. “Recent Developments in Models of Calcium Signalling.” Current Opinion in Systems Biology 3 (June 2017): 15–22. https://doi.org/10.1016/j.coisb.2017.03.002.

  1. Wacquier, Benjamin, Valérie Voorsluijs, Laurent Combettes, and Geneviève Dupont. “Coding and Decoding of Oscillatory Ca2+ Signals.” Seminars in Cell & Developmental Biology, SI: Calcium signalling, 94 (October 1, 2019): 11–19. https://doi.org/10.1016/j.semcdb.2019.01.008.

  1. Dupont, Geneviève, Laurent Combettes, Gary S. Bird, and James W. Putney. “Calcium Oscillations.” Cold Spring Harbor Perspectives in Biology 3, no. 3 (March 1, 2011): a004226. https://doi.org/10.1101/cshperspect.a004226.

- It is good to see a sensitivity analysis, but it should be discussed more. Also, given that the ODE model is quite cheap to simulate, a more comprehensive analysis would be desirable. The authors should also comment on why they do not change more than one parameter at the time.

Response:

We thank the Referee for this comment. We have now carried out a global sensitivity analysis and presented the results and discussion of the p-values and PRCC coefficients of the key drug and SOCE channel parameters in the modified version of the manuscript. Please see Table 3 and the associated discussion.

- The authors mention that they obtained some parameter values from fitting their model to experimental data. More details are needed. The only reference to a MATLAB function is not enough to understand the fitting procedure and the quality of the fit.

Response:

We thank the Referee for this comment. We have now provided the detail of the model fitting process in the manuscript as follows: To carry out the numerical simulations, the MATLAB’s ode23s built-in function, based on the explicit Runge-Kutta (2,3) pair of Bogacki and Shampine, was used to solve the systems of differential equations. The MATLAB’s lsqcurvefit built-in function was implemented using the Levenberg-Marquardt algorithm [30] that solves a minimization problem involving a least-squares data-fitting term for fitting the model to the experimental data presented in Table 1.

- A ratio of cytoplasmic volume to ER volume of 5.5 seems rather small. Could the authors provide a justification for this and put in into context with measured volume ratios?

Response: It has been reported that the ER can comprise >10% and cytosol can comprise 50-60% of the cellular volume. Therefore, we selected a ratio of cytoplasmic volume to ER volume at 5.5.

Ref.

http://book.bionumbers.org/how-big-is-the-endoplasmic-reticulum-of-cells/

- Figure 4 compares simulation results with experimental data. While the frequency of the oscillations is comparable, the shape of the oscillations does not match at all. While one could argue that the manuscript is primarily concerned with the frequency of the oscillations, the wave form matters (see e.g. Dupont, G., & Combettes, L. (2016). Fine tuning of cytosolic Ca2+ oscillations. F1000Research, 5. http://doi.org/10.12688/f1000research.8438.1). The reasons for the discrepancy between model and data needs to be discussed.

Response: We agree with the Reviewer’s comments. Both frequency modulation and shape/amplitude modulation of Ca2+ oscillations are important for downstream signaling transduction. For example, the frequency modulation controls NF-κB, MAPK, etc [24]. Our previous studies showed that the main downstream signaling pathway of Ca2+ oscillations in ESCC cells is NF-κB [14]. Therefore, in this study, we are primarily concerned with evaluating the anti-cancer effects of the combination drugs to control the frequency of oscillations. The results of this work serve as a motivation for a more extensive future study with stochastic dynamics to incorporate the effects of drugs on oscillations and other signaling components. We included the discussion of these points into the revised version (Line 434-453).